# The Silencing of *GhPIP5K2* and *GhPIP5K22* Weakens Abiotic Stress Tolerance in Upland Cotton (*Gossypium hirsutum*)

**DOI:** 10.3390/ijms25031511

**Published:** 2024-01-26

**Authors:** Pingjie Ling, Jisheng Ju, Xueli Zhang, Wei Wei, Jin Luo, Ying Li, Han Hai, Bowen Shang, Hongbo Cheng, Caixiang Wang, Xianliang Zhang, Junji Su

**Affiliations:** 1College of Life Science and Technology, Gansu Agricultural University, Lanzhou 730070, China; linger12012021@163.com (P.L.); 18893855307@163.com (J.J.); 15117242582@163.com (X.Z.); 17865557657@163.com (W.W.); 18992209045@163.com (J.L.); 17834316987@163.com (Y.L.); 13653802626@163.com (H.H.); 15872228546@163.com (B.S.); chenghbdd@126.com (H.C.); wangcaix@gsau.edu.cn (C.W.); 2State Key Laboratory of Cotton Biology, Institute of Cotton Research, Chinese Academy of Agricultural Sciences (CAAS), Anyang 455000, China; 3Western Research Institute, Chinese Academy of Agricultural Sciences (CAAS), Changji 831100, China

**Keywords:** cotton, *GhPIP5K2*, *GhPIP5K22*, abiotic stresses, VIGS

## Abstract

Phosphatidylinositol 4-phosphate 5-kinases (PIP5Ks), essential enzymes in the phosphatidylinositol signaling pathway, are crucial for the abiotic stress responses and the overall growth and development of plants. However, the *GhPIP5Ks* had not been systematically studied, and their function in upland cotton was unknown. This study identified a total of 28 *GhPIP5Ks*, and determined their chromosomal locations, gene structures, protein motifs and cis-acting elements via bioinformatics analysis. A quantitative real-time PCR (qRT‒PCR) analysis showed that most *GhPIP5Ks* were upregulated under different stresses. A virus-induced gene silencing (VIGS) assay indicated that the superoxide dismutase (SOD), peroxidase (POD), and catalase (CAT) activities were significantly decreased, while malondialdehyde (MDA) content were significantly increased in *GhPIP5K2*- and *GhPIP5K22*-silenced upland cotton plants under abiotic stress. Furthermore, the expression of the stress marker genes *GhHSFB2A*, *GhHSFB2B*, *GhDREB2A*, *GhDREB2C*, *GhRD20-1*, *GhRD29A*, *GhBIN2*, *GhCBL3*, *GhNHX1*, *GhPP2C*, *GhCBF1*, *GhSnRK2.6* and *GhCIPK6* was significantly decreased in the silenced plants after exposure to stress. These results revealed that the silencing of *GhPIP5K2* and *GhPIP5K22* weakened the tolerance to abiotic stresses. These discoveries provide a foundation for further inquiry into the actions of the *GhPIP5K* gene family in regulating the response and resistance mechanisms of cotton to abiotic stresses.

## 1. Introduction

Phosphatidylinositol (4,5)-bisphosphate [PtdIns (4,5) P2] is produced by the phosphorylation of phosphatidyl inositol phosphate 4 [PtdIns (4) P] or phosphatidyl inositol 5 phosphate [PtdIns (5) P] through the action of phosphatidyl inositol phosphate kinase (PIP kinase) [1] and plays a crucial role as a signaling molecule in salt and osmotic stress resistance [2], vesicle transport [3], actin tissue [4,5], and vanadate-sensitive H^+^-ATPase [6] to regulate the plasma membrane and ion channel activity [7]. PtdIns (4,5) P2 serves as both a lipid signal that interacts with effector proteins and a substrate for phospholipase C signaling. This signaling pathway regulates various cellular processes, including cytoskeletal organization and membrane trafficking [8,9,10,11,12,13,14], through signal transduction, guard cell movements [15] and pollen tube growth [16,17]. The relevance of the signaling function of PtdIns (4,5) P2 relied strongly on its spatiotemporal distribution, which was predominantly determined by its metabolic activity within specific membrane regions. The spatial-temporal configuration of PtdIns (4,5) P2 was determined primarily by the enzymatic processes involved in its synthesis and breakdown. Among these processes, the presence of phosphatidylinositol 4-phosphate 5-kinases (PIP5Ks), members of the *PIPK* family, played a crucial role in shaping the pattern of PtdIns (4,5) P2 [18,19,20,21,22]. Various isoenzymes of *PIP5K* have been detected [18,23,24,25], and mammals possess three distinct *PIP5K* isozymes that played crucial roles in diverse physiological processes, such as regulating the dynamics of the actin cytoskeleton, facilitating endocytosis and exocytosis, promoting cytokinesis, influencing apoptosis, and participating in nuclear processes [18,26,27,28,29,30,31,32]. These isozymes of *PIP5K* also made significant contributions in fungi. For instance, yeast Fab1p, which shared similarities with human type II *PIP5K*, had an impact on both vesicle function and morphology [33]. These results suggested that the *PIP5K* genes played an essential role in the physiological processes of animals and fungi.

Higher plant genomes encoded relatively more *PIP5Ks* than animal and fungal genome [34,35]. *PIP5K* genes had been identified in model plants such as *Arabidopsis thaliana* and rice [36,37]. A total of 11 *PIP5K* genes had been found in *A. thaliana*, and these genes exhibited high similarity to animal-derived type I PtdInsP kinases. These proteins were further classified into types A (*PIP5K10* and *PIP5K11*) and B (*PIP5K1-9*) based on their structural differences, namely, the presence or absence of membrane occupancy or a repeating MORN motif at the N-terminus. The nine genes of *A. thaliana* categorized as type B were classified into three subgroups based on their sequence similarity, which is functionally conserved, and these subgroups comprised *PIP5K1-3*, *PIP5K4-6* and *PIP5K7-9* [34].

In recent years, several studies had reported discoveries regarding the biological functions of certain *PIP5Ks.* For instance, the expression of *AtPIP5K1* in *A. thaliana* could be rapidly induced by external stimuli such as drought, salt, and abscisic acid (ABA) [36,38], and its regulation was also associated with soluble protein kinases [39]. *AtPIP5K1* and *AtPIP5K2* were involved in pollen development [36,40,41]. In *PIP5K1* and *PIP5K* mutant pollen grains, they significantly contributed to the formation of vacuoles and the development of pollen, resulting in defective vacuoles and compromised outer wall formation. *AtPIP5K4* and *AtPIP5K5* had been found to contribute to the facilitation of pollen germination and tube elongation [42]_._ *AtPIP5K4*, *AtPIP5K5* and *AtPIP5K6* were redundantly involved in pollen germination [43]. Furthermore, *AtPIP5K3* regulated the elongation of root hairs through specific expression in the roots [44]. *AtPIP5K9* interacted with the cytoplasmic enzyme CINV1 and thus exerted an inhibitory effect on root cell elongation via sugar regulation [45]. *AtPIP5K7*, *AtPIP5K8*, and *AtPIP5K9* redundantly participated in root growth under osmotic stress conditions [46]. A single gene, *OsPIP5K1*, had been found to play a crucial role in the heading process of rice [37]. The *PIP5K* genes involved in pollen development in wheat [47] and pepper [48], and overexpression of soybean *GmPIP5K* enhanced the drought stress tolerance of *A. thaliana* [49]. These results indicated that *PIP5Ks* played a crucial role in the adaptation of plants to abiotic conditions, and in their growth and development processes.

Cotton (*Gossypium* spp.) is an important source of natural renewable fiber, edible oil and proteins and holds immense economic value as one of the foremost global cash crops. The reproduction of cotton plants is easily affected by many types of unfavorable conditions, which could lead to low yields. Compared with those of other crops, cotton has better abiotic stress tolerance, but its yield under extreme temperature, salinity and drought stress conditions remains threatened. The *PIP5K* genes were crucial for regulating stress responses and pollen generation and served as a key component in signaling pathways. However, the exact role of *GhPIP5Ks* in the regulation of the stress response and growth of cotton plants had not been determined. Through a genetic analysis of upland cotton, we successfully identified 28 *GhPIP5Ks*. This paper reported the taxonomy, chromosome distribution and evolution of *GhPIP5Ks*. We employed a quantitative real-time PCR (qRT‒PCR) assay to assess the expression patterns of *GhPIP5Ks* and thus investigated their responses to various abiotic stressors. The virus-induced gene silencing (VIGS) technique was applied to silence the target gene, and various physiological and biochemical indicators and stress marker genes were then detected to validate the function of the target gene.

## 2. Results

### 2.1. Identification of GhPIP5K Genes and Their Biophysical Properties

In this study, we identified 28 *G. hirsutum* genes that possess uniform structural domains and these *GhPIP5Ks* were designated *GhPIP5K1-28* based on their chromosomal location in *G. hirsutum* (Table 1). The protein size of the *GhPIP5Ks* ranged from 384 (*GhPIP5K27*) to 825 (*GhPIP5K2* and *GhPIP5K16*) amino acids (AAs), and the molecular weights varied between 44,353.16 (*GhPIP5K27*) and 93,092.51 (*GhPIP5K2*). The aliphatic indices of the *GhPIP5Ks* ranged from 59.79 (*GhPIP5K24*) to 86.58 (*GhPIP5K13*), and the protein instability indices ranged from 28.89 (*GhPIP5K12*) to 55.42 (*GhPIP5K25*). Additionally, the aliphatic indices of the *GhPIP5Ks* varied between 59.79 (*GhPIP5K9* and *GhPIP5K24*) and 86.58 (*GhPIP5K13*) and the grand average of hydropathicity was negative. The subcellular localization of 28 *GhPIP5Ks* were predicted, and the genes of this family were found to be localized mainly in the nucleus. Among the *GhPIP5K* genes, 78.57% were found to be present in the nucleus (22), 17.86% *GhPIP5Ks* were localized in the cytosol (5), and only one gene (*GhPIP5K24*) was found in the chloroplast. These findings suggest that *GhPIP5Ks* are predominantly expressed in the nucleus and cytosol.

### 2.2. Phylogenetic Analysis of the GhPIP5K Genes 

To explore the evolutionary connections among *PIP5Ks* derived from seven plant species, we constructed a phylogenetic tree (Figure 1) with the PIP5K proteins from *G. raimondii* (14), *G. arboreum* (14), *G. hirsutum* (28), *A. thaliana* (11), *Z. mays* (9), *O. sativa* (10) *and T. cacao* (12) (Appendix A). According to the phylogenetic tree of *PIP5K* genes, 98 *PIP5Ks* were clustered into three groups, namely, I, II and III, and group III was unevenly subdivided into three subgroups, namely, III-1, III-2 and III-3. Group III was the largest clade, with 24 *GhPIP5Ks* from *G. hirsutum*, whereas group II had only 4 members, namely, *GhPIP5K11*, *GhPIP5K13*, *GhPIP5K25* and *GhPIP5K27*. Group I lacked *GhPIP5Ks* and consisted of only two genes, *OsPIP5K7* and *OsPIP5K8*. The *PIP5K* genes originated from both dicotyledonous and monocotyledonous species. Some of the *PIP5K* genes in the III-3 subgroup, were derived from monocotyledonous plants, including five genes originating from maize and five genes originating from rice. The *PIP5K* genes in monocotyledonous plants (rice and maize) tended to cluster together, whereas those in dicotyledonous plants (*A. thaliana*, *T. cacao* and *Gossypium*) exhibited comparable characteristics. These findings indicated that the distribution of *PIP5K* members in *G. hirsutum* varied among the different groups. In addition, the phylogenetic tree showed that the *PIP5K* genes underwent a series of genomic amplification events during evolution from diploid to tetraploid cotton, which resulted in causing the allotetraploid cotton species *G. hirsutum* having twice as many *PIP5K* genes as *G. arboreum* and *G. raimondii.*

### 2.3. Chromosome Localization and Duplication Analysis of GhPIP5Ks

To ascertain the chromosomal positions of the *GhPIP5Ks* in *G. hirsutum*, we examined their physical distribution across chromosomes utilizing location data files obtained from the Cotton Functional Genomics Database (CottonFGD) website. The 28 *GhPIP5Ks* were unevenly distributed on the chromosomes (Figure 2). The 28 *GhPIP5Ks* were deposited on 18 homologous chromosomes, except for A06, A07, A09, A11, D06, D07, D09 and D11. Among all the chromosomes, chromosome A05 had the highest number of *GhPIP5Ks*, with a total of 4. In contrast, only one *GhPIP5K* was found on each of the following chromosomes: A02, A03, A08, A10, A12, A13, D02, D03, D08, D10, D12 and D13. Two or three *GhPIP5Ks* were found on the other chromosomes.

We analyzed the duplication patterns of 28 *GhPIP5Ks*, and 41 homologous duplicated gene pairs were found (Figure 3a). Among these replicated gene pairs, 3 formed between chromosomes A05 and D05 (*GhPIP5K7/8*, *GhPIP5K9/10* and *GhPIP5K22/23*) by tandem replication and the other 39 pairs originated from fragment replication. These findings indicated that the evolution of *GhPIP5Ks* were significantly influenced by the occurrence of segment repetition and tandem replication and that segment repetition played a dominant role relative to tandem replication in the development of *GhPIP5Ks*. Furthermore, to improve the understanding of the functional and evolutionary relationships of *PIP5Ks*, we performed a comparative analysis of the intergenomic synteny between *G. hirsutum* and two additional cotton species (Figure 3b). A collinearity analysis revealed 165 pairs of genes that exhibited collinearity between *G. hirsutum* and both *G. arboreum* and *G. raimondii*. These findings suggested that genomic rearrangements might have occurred in *PIP5K* genes during polyploidy events. To understand the factors that shape the differentiation of *GhPIP5Ks*, we calculated the nonsynonymous substitution rate (Ka), synonymous substitution rate (Ks), and nonsynonymous to synonymous substitution ratio (Ka/Ks). Interestingly, all pairs of duplicated *GhPIP5Ks* had Ka/Ks ratios lower than 1, indicating the occurrence of strong purifying selection on these gene pairs (Appendix A).

### 2.4. Gene Structure and Conserved Motifs of GhPIP5K Proteins

To explore the structural diversity of the GhPIP5K proteins, their conserved patterns were examined. A total of 10 motifs in the *GhPIP5Ks* were labeled motifs 1–10. The number of these conserved motifs varied among different GhPIP5K proteins, and the motif compositions tend to be similar among members of the same subfamily, suggested functional differentiation among GhPIP5K proteins (Figure 4b).

To comprehensively study the similarity and diversity of the GhPIP5K proteins, we explored the variation and evolutionary process of the structure of genes within the *PIIP5K* family in upland cotton through a detailed investigation of introns and exons. The analysis revealed differences in the lengths of the *GhPIP5Ks*. Specifically, *GhPIP5K5* had the longest genome sequence (11.88 kb), whereas *GhPIP5K11* had the shortest (Figure 4c). Within the same subclass, most *GhPIP5Ks* exhibited similar gene structure characteristics regarding the number and length of introns and exons (Figure 4a). These findings indicated that the *GhPIP5Ks* exhibit significant conservation throughout plant evolution and potentially played analogous roles in both plant development and defense mechanisms.

### 2.5. Exploration of the Regulatory Elements within the Promoter Regions of GhPIP5K Genes

The regulation of downstream genes was significantly influenced by the presence of cis-acting elements in the initiator region of genes. To gain insight into the role of the *GhPIP5Ks*, we obtained cis-acting elements from the 5’-upstream region 2000 bp of each gene. We categorized the cis-acting elements in the *GhPIP5K* promoter into four groups, namely, hormone-responsive, light-responsive, abiotic stress-responsive, and growth and development-related promoters (Figure 5). As part of the abiotic stress response, various elements, including ARE (84), GC-motif (6), LTR (17), MBS (14), TC-rich repeats (10) and the WUN-motif (29), had been detected in *G. hirsutum*. These findings suggested that *GhPIP5Ks* might participate in diverse regulatory mechanisms in cotton plants under various stress conditions, providing clues for the selection of candidate genes for subsequent experiments. Many light-responsive elements were also detected in these *GhPIP5Ks*. Among these genes, box 4 was the most common detected light-responsive element and exhibited the highest frequency 107 in 28 promoters of *GhPIP5K*. Various elements, such as ABA response element (ABRE), the CGTCA and TGACG motifs (MeJA-responsive element), the P-box, the TATC-box and GARE-motif (GA-responsive element), and the TCA-element (salicylic acid-responsive element), exhibited responsiveness to phytohormones. These findings suggested that the regulation of *GhPIP5K* expression was influenced by a multitude of plant hormones. Other identified cis-regulatory elements implicated in plant growth and development included RY elements (3), which were associated with seed-specific regulation; O2 sites (13), which were involved in metabolic modulation; CAT boxes (8), which were linked to meristem expression; and GCN4 motifs (3), which participated in endosperm expression. These findings suggested that *GhPIP5Ks* might play essential roles in plant growth and development.

### 2.6. Expression Profiling of GhPIP5K Genes in G. hirsutum

To investigate the expression patterns of *GhPIP5Ks* in distinct upland cotton tissues, we examined the expression levels of *GhPIP5Ks* using transcriptome data with the aim of exploring their biological functions. The analysis included a range of tissues, such as roots, stems, leaves, torus, petals, sepals, bracts, anthers, filaments and pistils. Additionally, we examined various developmental stages, ranging from 3 days before to 5 days after anthesis (dpa), as well as fibers and ovules at 10–25 dpa (Appendix A). We found that the *GhPIP5Ks* in subfamily III were expressed mainly in floral organs and flowering-related tissues. Among them, *GhPIP5K1*, *GhPIP5K10*, *GhPIP5K14*, *GhPIP5K24*, *GhPIP5K25*, *GhPIP5K26*, *GhPIP5K27* and *GhPIP5K28* were expressed mainly during anthesis, whereas *GhPIP5K5* and *GhPIP5K13* were expressed mostly in the roots. Other genes were found to be expressed in various tissues. The data indicated that *GhPIP5Ks* were not only involved in controlling the growth of both the above and belowground parts sections of cotton but also had an impact on its reproductive processes.

To verify the RNA-seq data, we conducted a qRT‒PCR analysis of eight specific *PIP5K* genes derived from *G. hirsutum*, which allowed us to examine the expression patterns of these genes in various plant tissues, including roots, stems, leaves, bracts, petals, sepals, pistils, stamens and fibers (Figure 6). The results showed that six genes (*GhPIP5K2*, *GhPIP5K6*, *GhPIP5K10*, *GhPIP5K15*, *GhPIP5K17* and *GhPIP5K25*) exhibited increased expression in the stamens. The petals displayed notable increases in the expression of three genes, namely, *GhPIP5K7*, *GhPIP5K15*, and *GhPIP5K22*. High expression levels of *GhPIP5K6*, *GhPIP5K15* and *GhPIP5K17* high were also detected in root tissues. These findings suggested that *GhPIP5Ks* are strongly linked to floral organ development, especially in stamen tissue. The tissue expression levels of 8 specific genes in *G. hirsutum* were investigated by qRT‒PCR analysis, and the results exhibited a high degree of concordance with the RNA-seq data.

### 2.7. Verification of the Response of GhPIP5Ks to Four Types of Abiotic Stress by qRT‒PCR

Considering the functional roles of genes under different environmental constraints, we conducted a transcriptome analysis of 28 *GhPIP5Ks* under stress treatments (PEG, NaCl, and high and low temperature) (Appendix A). The results revealed that the various *GhPIP5Ks* exhibited different responses to these stress treatments. In the present research, we examined the expression profiles of *GhPIP5Ks* in response to four abiotic stress conditions, taking XinshiK25 as the research object. We used qRT‒PCR to determine whether the *PIP5K* genes were involved in the abiotic stress response of cotton. Eight genes were selected for analysis of their expression in materials treated with salt, drought, heat and cold (Figure 7). Compared with that in the control, the expression of seven genes in the drought stress, treatment was significantly increased; for example, the expression of *GhPIP5K2* was significantly increased by hundreds of times, whereas the expression of *GhPIP5K25* was significantly decreased. Under NaCl stress, the expression of *GhPIP5K2*, *GhPIP5K6* and *GhPIP5K17* increased exponentially, peaking at 12 h, whereas the expression of *GhPIP5K25* significantly decreased, and the expression of the other genes did not significantly change. Heat stress significantly increased the expression of *GhPIP5K2*, *GhPIP5K10*, *GhPIP5K17*, *GhPIP5K22* and *GhPIP5K25* and significantly decreased the expression of *GhPIP5K7*. In addition, the expression of *GhPIP5K6* and *GhPIP5K15* first increased and then decreased, peaking at 1 h. With the exception that of *GhPIP5K7* and *GhPIP5K15*, the expression of the other 6 genes was significantly upregulated under cold stress.

In general, the expression of *GhPIP5K2* increased hundreds of times under drought, NaCl and heat stress, and tens of times under cold stress. Similarly, the expression of *GhPIP5K17* increased approximately tenfold under the four abiotic stresses. These findings indicated that these two genes are broadly related to four abiotic stresses and could be regarded as potential candidate genes for improving the stress tolerance of cotton. The expression of *GhPIP5K22*, significantly increased hundreds of times under heat stress and cold stress conditions. Similarly, the expression of *GhPIP5K15* was significantly upregulated, especially under cold stress conditions. These results showed that these two genes strongly respond to temperature changes and can be used as candidate genes for improving the tolerance of cotton. 

### 2.8. The Silencing of GhPIP5K2 and GhPIP5K22 Compromises the Tolerance of Cotton to Stress

In addition, we preliminarily explored the function of *GhPIP5K2* under four stress conditions (high and low temperature, drought and NaCl) and *GhPIP5K22* under two stress conditions (high and low temperature) by the VIGS technique. Nine days after infection, the cotton leaves exhibited a phenotype characterized by an albino appearance (Figure 8a). TRV:*GhCLA1* was used as the positive control to validate the efficacy of the VIGS technique. We then analyzed the function of *GhPIP5K2* under high temperature (42 °C), low temperature (12 °C), drought and NaCl (200 mmol/L) conditions and that of *GhPIP5K22* under high temperature and low temperature treatments. The silenced and normal plants were compared, and the results showed that the plants in which the two target genes were silenced exhibited characteristics such as wilting, yellowing of leaves, and lack of water (Figure 8c,d and Figure 9e). The new leaves of the TRV:*GhPIP5K2*-silenced plants exhibited blackening under cold stress compared with those of the control plants (Figure 8d). These results indicated that *GhPIP5K2* had a positive effect on the responses of cotton to abiotic stress, whereas *GhPIP5K22* had a positive effect on the responses to heat and cold conditions.

To explore the impact of abiotic stress on *GhPIP5K2* and *GhPIP5K22*, we assessed the variations in superoxide dismutase (SOD), peroxidase (POD), and catalase (CAT) activities and malondialdehyde (MDA) content in the TRV:*GhPIP5K2*- and TRV:*GhPIP5K22*-silenced cotton plants. Our findings revealed significant decreases in antioxidant enzyme activities (SOD, POD, and CAT) in the silenced plants compared with the TRV:00 plants. Conversely, a notable increase in the MDA content was detected (Figure 10). The above-described findings suggested that the silencing of *GhPIP5K2* and *GhPIP5K22* impaired the tolerance of cotton plants to abiotic stress. In addition, compared with the control plants, the TRV:*GhPIP5K2* and TRV:*GhPIP5K22* plants exhibited significant changes in the expression of stress-related genes before and after stress treatment. Notably, the expression of *GhBIN2*, *GhCBL3* and *GhNHX1* was upregulated, in the TRV:*GhPIP5K2* plants and significantly downregulated in the silenced plants following stress treatment. The expression levels of *GhDREB2A*, *GhHSFB2C*, *GhRD20-1*, *GhPP2C* and *GhSnRK2.6* were lower in the silenced plants than in the control plants both before and after treatment. Similarly, the expression of *GhHSFB2A*, *GhDR29A* and *GhCBF1* was downregulated in the silenced plants compared with the control plants before treatment, whereas the expression of these genes in the silenced plants after at least one stress treatment was significantly lower than that in the control plants (Figure 11a). These results suggested that *GhPIP5K2* might positively regulate abiotic stress. *GhHSFB2B* expression was upregulated in the TRV:*GhPIP5K22* (control) plants, but was significantly downregulated in the silenced plants following high-temperature stress treatment. Three other stress-related genes (*GhDREB2A*, *GhDREB2C and GhRD29A*) exhibited similar patterns. Conversely, the expression levels of *GhRD20-1*, *GhCIPK6* and *GhCBF1* were lower in the silenced plants than in the control plants both before and after treatment (Figure 11b). These findings suggested the direct impact of the silencing of *GhPIP5K22* on the expression levels of *GhRD20-1*, *GhCIPK6* and *GhCBF1*. In summary, we proposed that *GhPIP5K22* might play a positive regulatory role during the response to temperature stress.

## 3. Discussion

PIP5Ks are phosphor ester kinases, and their numbers vary among different plants. *PIP5K* genes have been discovered in various dicotyledonous crop species, such as *A. thaliana* [34], *Glycine max* [35], and *Capsicum annuum* [48], which contain 11, 22, and 19 genes, respectively. Additionally, these genes had been found in monocotyledonous crops such as *O. sativa*, which contains a total of 10 genes [35]. Fifty-six *PIP5Ks* were identified in *G. hirsutum* (28), *G. raimondii* (14) and *G. arboreum* (14) in this study. Interestingly, the number of *GhPIP5Ks* were the aggregate of the amounts of *GaPIP5K* and *GrPIP5K* genes, probably because *G. hirsutum* were a heterotetraploid crop produced by crossing the ancestors of the A and D genomes. In upland cotton, both A_t_ and D_t_ subgenomic donors were directly homozygous, leading to duplication of the *GhPIP5Ks* [50]. The replication and amplification of *GhPIP5Ks* had led to a higher abundance of *GhPIP5Ks* compared with that of the *GaPIP5K* and *GrPIP5K* genes. Nearly twice as many genes had been detected in the tetraploid plants *G. hirsutum* and *G. max* as in the diploid plants *A. thaliana* and *O. sativa*.

An in-depth study was conducted to identify the *PIP5Ks* across the cotton genome, and the evolutionary relationships among *G. hirsutum*, two other cotton species, *A. thaliana*, *maize*, *O. sativa*, and *T. cacao*, were then analyzed. Based on the phylogenetic analysis, PIP5K proteins could be classified into three main categories: I, II, and III. Moreover, group III could be divided into three distinct subgroups, namely, III-1, III-2, and III-3, which were formed by 98 genes. Notably, Figure 2 and Figure 5 illustrated the conserved characteristics of each subgroup, including motifs, gene structures, and domain features. These findings suggested a potential correlation between these groups and their involvement in plant growth and developmental processes. The majority of the *GhPIP5Ks* in group III exhibited structural domains such as MORN repeat motifs and PIPKc. The origin of the MORN structural domain in plants could be traced back to a primitive protein. Notably, the MORN structural domains found in plant PIP5K proteins exhibit significant dissimilarities compared with those observed in other protein variants [51]. The SMART database revealed that all 13 members of group I and II possess typical PIPKc domains, indicated that the gene architecture and domain characteristics exhibit a consistent pattern within the same group.

Duplicated genes serve as the fundamental building blocks for generating novel genes, thereby enabling the emergence of new functionalities. Gene duplication was widely recognized as a major contributor to the proliferation of plant gene families, and fragment replication and tandem replication had been identified as the primary mechanisms driving this expansion [52]. In this study, we investigated the correlation of *GhPIP5Ks* with chromosomes in upland cotton and noted that segment and tandem duplications were responsible for the amplification of the *PIP5K* family. Tandem duplication events between the genomes were found on chromosomes A05 and D05. Chromosomal fragment duplication events occurred more frequently in different genomes. Segment duplications of *GhPIP5Ks* were more abundant in *G. hirsutum* than in *G. max* [35]. However, in heterozygous hexaploid wheat, only fragment replication events occurred, with no tandem duplication [35]. The analysis of Ka/Ks ratios indicated that the PIP5K protein family underwent selective purification, indicated its stability throughout long-term evolution.

The presence of *PIP5Ks* had been documented in diverse dicotyledonous crop species, such as *A. thaliana*, *G. max* and *C. annuum*, and in monocot species, such as wheat and rice. Among these species, these genes had most widely been reported in *A. thaliana* and the *PIP5K* genes in *A. thaliana* were previously demonstrated to play a crucial role in various physiological processes. Specifically, these genes govern lateral root growth and hair tip elongation, facilitate stomatal opening, regulated then intracellular calcium ion levels, respond to water stress, and participated in the regulatory pathway for ABA signaling. In the context of the plant stress response, the involvement of cis-regulatory elements at the promoter site was crucial [53]. In this study, a promoter assay revealed that *GhPIP5K* contains cis-acting elements that respond to phytohormones and abiotic stresses, such as ABA and low temperature, suggested a potential role for *GhPIP5K* in modulating diverse responses to plant hormones, environmental stresses and development. These findings were comparable to the outcomes obtained from a gene promoter analysis of maize *PIP5Ks* [54]. In addition, most of the *GhPIP5Ks* contained growth- and development-related elements and light-responsive elements, which was consistent with the findings from previous studies of pepper [48]. Several studies had investigated stress-related *PIP5Ks*. In *A. thaliana*, the genes *AtPIP5K1*, *AtPIP5K7*, *AtPIP5K8* and *AtPIP5K9* were reportedly associated with salt and drought stress. In this study, we found that *GhPIP5K7*, *GhPIP5K22* and *AtPIP5K1*; *GhPIP5K2*, *GhPIP5K6*, *GhPIP5K17* and *AtPIP5K7*, *AtPIP5K8* and *AtPIP5K9* were clustered in the same subgroups in the evolutionary classification tree. Thus, we predicted that these five genes in *G. hirsutum* also had functions related to the responses to abiotic stresses. An examination of gene expression models could be used for the prediction of gene functions [55]. The expression patterns of *GhPIP5Ks* during diverse tissue development stages and under abiotic stress were investigated based on RNA-seq data and real-time fluorescence quantitative preliminary verification in upland cotton. A heatmap revealed that the expression of *GhPIP5K2*, *GhPIP5K6*, *GhPIP5K7*, *GhPIP5K10*, *GhPIP5K15*, *GhPIP5K17*, *GhPIP5K22* and *GhPIP5K25* changed under salt stress, drought stress, heat stress and cold stress conditions. These predictions were also verified by real-time fluorescence quantification. Among these genes, the *GhPIP5K2* and *GhPIP5K17* exhibited the most significant changes in expression under drought, salt, heat and cold stresses, whereas *GhPIP5K22* exhibited the most significant changes in expression under temperature stress. These three genes could serve as candidates for future research on gene editing for breeding applications.

Upon exposure to stress, plants undergo a rapid increase in their intracellular levels of reactive oxygen species (ROS), which subsequently disrupts the normal physiological and metabolic functions of plant cells [56]. The synthesis of SOD, POD and CAT enzymes serves as an indispensable protective mechanism against the deleterious effects induced by harsh environmental conditions because these enzymes effectively scavenge ROS within plant cells. To further explored the impact of *GhPIP5Ks* on cotton plants in response to abiotic stress, we silenced *GhPIP5K2* and *GhPIP5K22* using VIGS technology. Our findings showed that knock down of the target gene resulted in heightened sensitivity to drought and temperature and accelerated the yellowing and wilting of leaves in cotton plants. We then assessed the levels of SOD, POD and CAT activities and the MDA content in response to exposure to abiotic stress. Taken together, our findings suggested that the TRV:*GhPIP5K2* and TRV:*GhPIP5K22* plants exhibited lower SOD, POD and CAT activities and a higher MDA content than the negative control plants. These findings indicated that *GhPIP5K2* and *GhPIP5K22* could modulate the adaptation to abiotic stress by augmenting the ability of cotton to eliminate ROS. Previous research had yielded comparable findings [57]. We further tested the expression levels of stress marker genes (*GhHSFB2A*, *GhHSFB2B*, *GhDREB2A*, *GhDREB2C*, *GhRD20-1*, *GhRD29A*, *GhBIN2*, *GhCBL3*, *GhNHX1*, *GhPP2C*, *GhCBF1* and *GhCIPK6*) in the negative control and silenced plants and found significant decreases in their expression levels in the silenced plants in response to exposure to at least one stress condition [58,59,60]. These findings strongly suggested that suppression of the target genes decreased the tolerance of plants to abiotic stress. Based on the abovementioned findings, it could be inferred that *GhPIP5K2* and *GhPIP5K22* played positive regulatory roles in regulating the responses of cotton to abiotic stress. In summary, as proteins with positive regulatory effects, *GhPIP5K2* and *GhPIP5K22* could potentially enhance the ability of terrestrial cotton ability to withstand abiotic stress by promoting stress-related gene expression and increasing the CAT, POD, and SOD enzyme activities. Specifically, after further VIGS validation, the findings obtained for *GhPIP5K2* and *GhPIP5K22s* were consistent with earlier predictions that these genes positively regulated the response to abiotic stress. Further comprehensive investigations and analytical tests were important for improving the agricultural production and stress tolerance of cotton plants.

## 4. Materials and Methods

### 4.1. Identification of Members of the PIP5K Gene Family 

We obtained genomic sequence and annotated data for three cotton species, namely *Gossypium hirsutum*, *Gossypium arboretum* and *Gossypium raimondii* from the CottonFGD [61] (https://cottonfgd.org/, accessed on 10 March 2023). Eleven *AtPIP5Ks* of *A. thaliana* were retrieved from the *A. thaliana* Information Resource [62] (TAIR, http://www.arabidopsis.org, accessed on 10 March 2023). One of these eleven AtPIP5K proteins was used to obtain the hidden Markov model profile (PF01504) of the *AtPIP5K* structural feature sequence from Pfam online software [63] (http://pfam.xfam.org/, accessed on 11 March 2023). We then utilized HMMER 3.0 software to retrieve the cotton protein sequence with the seed file as the index, set the E-value cutoff to 1.0 × 10^−5^ to ensure confidence, and obtained the sequence matching the structural characteristics of the protein, which was the candidate member of the *PIP5K* gene family. The protein sequences of all candidate *PIP5Ks* were analyzed using SMART [64] (https://smart.embl.de/, accessed on 11 March 2023) and NCBI Batch CD-Search [65] (https://www.ncbi.nlm.nih.gov/Structure/cdd/wrpsb.cgi, accessed on 11 March 2023) to identify their respective domains. The GhPIP5K proteins in upland cotton were analyzed using ExPASy [66] (https://web.expasy.org/protparam/, accessed on 12 March 2023), an online tool. The *GhPIP5Ks* were predicted using a subcellular localization site called WoLF PSORT [67] (https://wolfpsort.hgc.jp/, accessed on 13 March 2023) to determine their location in the cell.

### 4.2. Chromosomal Localization and Collinearity Analysis

Chromosome location information for the *GhPIP5Ks* were obtained by downloading the gene annotation files from the CottonFGD website [61] (https://cottonfgd.org/, accessed on 10 March 2023). An analysis of the spatial arrangement of the *GhPIP5Ks* across chromosomes were conducted with TBtools v1.116 software [68]. We used whole-genome sequences and gene annotations of *G. hirsutum*, *G. raimondii* and *G. arboreum* to pinpoint tandem and segmented replication events in *PIP5K* genes. The PIP5K proteins were subjected to multiple sequence analysis using MCScanX. TBtools software was subsequently applied to display the multicollinearity of the repeating genes and thus determined their relationships.

### 4.3. Sequence Alignments and Phylogenetic Analysis of the PIP5K Genes

We performed multiple sequence comparisons of PIP5K protein sequences from three cotton varieties using DNAMan 2.0 software. To analyze the phylogenetic relationships, we compared the protein sequences of four species (*Arabidopsis*, *Zea mays*, *Oryza sativa* and *Theobroma cacao*) and three cotton varieties (*G. hirsutum*, *G. raimondii* and *G. arboretum*). The MEGA 11.0 [69] algorithm was used to search for the optimal model and to construct a tree via the neighbor-joining (NJ) method. The tree was subjected to 1000 iterations using the bootstrap method. The Poisson model with default parameters was employed to determine substitutions.

### 4.4. Analysis of the Gene Structure, Conserved Motifs and Cis-Acting Elements 

To enhance our understanding of the conservation of *PIP5K* genes, we predicted the gene structures using the Gene Structure Display Server 2.0 [70] (http://gsds.cbi.pku.edu.cn/, accessed on 14 March 2023) online tool. The conserved motifs in the *PIP5K* genes were analyzed using the MEME database [71] (http://meme-suite.org/, accessed on 14 March 2023), with the selection of 10 motifs as parameters. We utilized the Gtf/Gff3 sequence extraction tool (Gtf/Gff3 sequence extraction) within TBtools software to retrieve a 2000bp promoter sequence located upstream of the coding DNA sequence for each *GhPIP5Ks*. These extracted sequences were subsequently subjected to cis-element prediction using PlantCare [72] (http://bioinformatics.psb.ugent.be/webtools/plantcare/html/, accessed on 15 March 2023).

### 4.5. Transcriptome Analysis of PIP5K Genes during Growth and Development and under Abiotic Stress Conditions

The expression patterns of the *GhPIP5Ks* were studied by obtaining RNA-seq data of TM-1 tissue and abiotic stress (ZJU)-treated plants from the cotton website (http://cotton.zju.edu.cn/, accessed on 17 March 2023). The TM-1 RNA-seq dataset encompasses diverse developmental phases, including those involving roots, stems, leaves, tori, petals, anthers and pistils. Additionally, the dataset included developmental stages: 0 to 20 dpa-ovule and 10 to 25 dpa-fibers. Furthermore, the dataset encompassed abiotic stress treatments involving salt, drought, cold and heat conditions. Heatmaps of all 28 *GhPIP5Ks* were generated using TBtools software.

### 4.6. Experimental Materials and Treatments

The upland cotton variety XinshiK25 was subjected to different abiotic stress treatments. Each small pot was filled with an equal amount of nutrient soil (substrate:vermiculite = 1:1) and placed in large pots filled with tap water for immersion until the water was absorbed to the surface of the pots to promote cottonseed germination. Seeds of Neolith K25 were planted in the abovementioned pots, and the depth of seed planting was maintained at 1.5 cm to achieve consistent seedling emergence. The plants were cultivated in an artificial climate incubator (16 h light, 8 h dark, 28 °C temperature, 70% humidity). Once the cotton plants had grown to the four-leaf stage, they were treated with either natural water loss or 200 mmol/L NaCl or subjected to a temperature of 12°C or 42 °C. Leaf samples were collected after treatment for 0, 1, 3, 6, 12, and 24 h and stored at −80 °C. The roots, stems, leaves, bracts, petals, sepals, pistils, stamens and fibers of the normally growing experimental material XinshiK25 were collected, and the samples were first rapidly frozen in liquid nitrogen and subsequently stored at −80 °C for later use.

### 4.7. Extraction of RNA and Quantitative Real-Time Polymerase Chain Reaction (qRT‒PCR) Analysis

RNA was isolated from the apical meristem and juvenile foliage of the plants and from nine distinct organs using a polysaccharide-polyphenol-based RNA extraction kit (Tiangen, Tianjin, China). A NanoDrop 2000 spectrophotometer (Thermo Scientific, Waltham, MA, USA) was used to assess the RNA quality of all the samples. The specimens were cryogenically preserved and maintained at a temperature of −80 °C. The RNA was then used as the template for generating cDNA through reverse transcription via the First-Strand cDNA Synthesis Kit (Tiangen, China). The qRT‒PCR primers (Appendix A) for the *PIP5K* genes were designed using Primer-BLAST (https://www.ncbi.nlm.nih.gov/tools/primer-blast/index, accessed on 19 March 2023). The internal control gene β-actin was utilized for housekeeping purposes. The real-time PCR experiment was performed with a 20 µL reaction mixture comprising 2.4 µL of each primer (at a concentration of 2.5 µM), 2 µL of cDNA (100 ng/µL), 10 µL of SYBR Premix Ex Taq (at a concentration of 2×), and 3.2 µL of ddH_2_O. The experiment was performed in triplicate with a LightCycler^®^ 96 (Roche, Switzerland, Europe) instrument. The reaction procedure involved incubation at 95 °C for 3 min followed by 40 cycles of 95 °C for 5 s and a final incubation at 60 °C for 15 s.

### 4.8. Silencing of GhPIP5K2 and GhPIP5K22 in Cotton 

The silencing of *GhPIP5K2* and *GhPIP5K22* was achieved via the VIGS technique using a TRV vector. Initially, 417-bp and 431-bp target fragments of *GhPIP5K2* and *GhPIP5K22* were acquired through PCR amplification and subsequently integrated into the TRV (PYL156) vector (Appendix A). The resulting construct was subsequently introduced into Agrobacterium strain GV3101 via the freeze‒thaw technique. The pYL192 strains were then mixed with the TRV:00, TRV:*GhCLA1*, TRV:*GhPIP5K2* and TRV:*GhPIP5K22* strained at a ratio of 1:1 and injected into 8-day-old cotton cotyledons after treatment under dark conditions for 3 h. After 24 h of cultivation in the dark at 25 °C, normal cultivation was performed. After the occurrence of positive control albinism, samples were collected and stored at −80 °C. The gene expression levels in TRV:*GhPIP5K2*- and TRV:*GhPIP5K22*-silenced plants were detected via real-time fluorescence quantitative PCR. TRV:00 empty carrier plants and TRV:*GhPIP5K2* plants were grown to 4 weeks of age and then subjected to high temperature (42 °C), low temperature (12 °C), drought (dehydration) and NaCl (200 mmol/L) conditions. Moreover, TRV:*GhPIP5K22* plants were treated with high or low temperature. After 10 days of stress treatment, the activities of antioxidant enzymes (CAT, POD, and SOD) and the content of MDA were determined. The expression levels of marker genes (*GhHSFB2A*, *GhDREB2A*, *GhDREB2C*, *GhRD20-1*, *GhRD29A*, *GhBIN2*, *GhCBL3*, *GhNHX1*, *GhPP2C*, *GhSnRK2.6* and *GhCBF1*) in the TRV:*GhPIP5K2*-silenced plants were measured before and after stress treatment, and the expression levels of marker genes (*GhHSFB2B*, *GhDREB2A*, *GhDREB2C*, *GhRD20-1*, *GhRD29A*, *GhCBF1* and *GhCIPK6*) in TRV:*GhPIP5K22*-silenced plants were measured before and after stress treatments.

## 5. Conclusions

We identified a total of 14, 14, and 28 *PIP5Ks* in *G. arboreum*, *G. raimondii*, and *G. hirsutum*, respectively, based on their genomic information. We examined the chromosomal distribution, gene structure, duplication events, conserved genes, cis-elements and expression patterns of the *GhPIP5Ks*. Notably, our expression profiling revealed the substantial involvement of the *GhPIP5Ks* in both the abiotic stress response and pollen growth regulation. Furthermore, preliminary confirmation through VIGS suggested that *GhPIP5K2* specifically contributes to abiotic stress tolerance and that *GhPIP5K22* was specifically involved in tolerance to temperature stress. Considering these findings, *GhPIP5K2* and *GhPIP5K22* are particularly desirable candidates for further exploration and utilization for augmenting the resistance of cotton against various stresses.

## Figures and Tables

**Figure 1 ijms-25-01511-f001:**
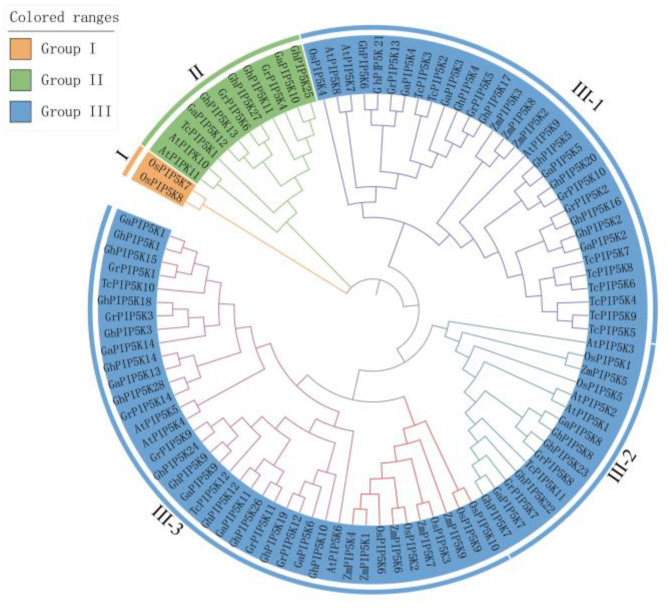
Evolutionary relationships among *PIP5K* genes. The sequences of *PIP5K* in *Arabidopsis thaliana*, *Zea mays*, *Oryza sativa*, *Theobroma cacao*, *G. arboretum*, *G. raimondii* and *G. hirsutum* were arranged using ClustalW and a phylogenetic tree was constructed through the neighbor-joining method with the aid of iTOL. The evolutionary tree was divided into three groups, and each of the three groups is represented by a different color: orange, group I; green, group II; and blue, group III.

**Figure 2 ijms-25-01511-f002:**
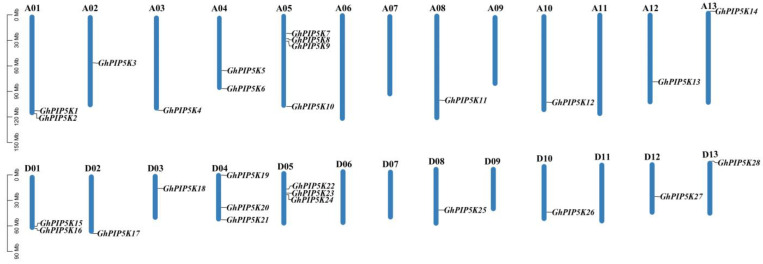
Locations of *GhPIP5Ks* on chromosomes in *G. hirsutum*. The blue bars represent chromosomes in the graph corresponding to the A_t_ and D_t_ subgenomes of *G. hirsutum*. The gene names corresponding to each chromosome can be observed on the right side of both the A_t_ and D_t_ subgenomes.

**Figure 3 ijms-25-01511-f003:**
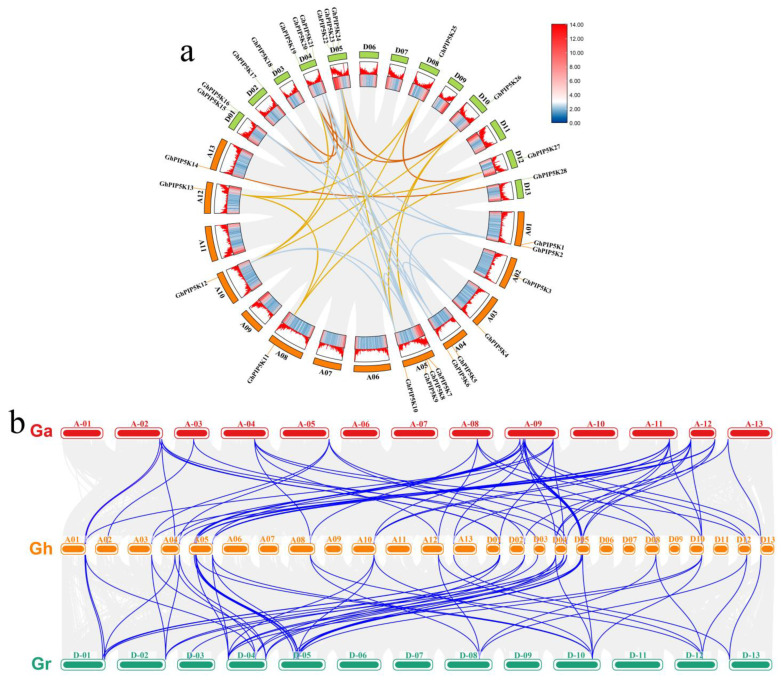
Duplication analysis of *PIP5Ks* within and between cotton species. (**a**). Duplication of *GhPIP5Ks* on the chromosome. The interconnections between all the genes in the genome of upland cotton were denoted by gray lines, and the *GhPIP5K* gene pairs were depicted by lines of different colors. Chromosomes were indicated by distinctly colored rectangles. (**b**). Collinearity analysis diagram of three cotton species; *G. arboreum* is shown in red, *G. hirsutum* is shown in orange, and *G. raimondii* is shown in cyan.

**Figure 4 ijms-25-01511-f004:**
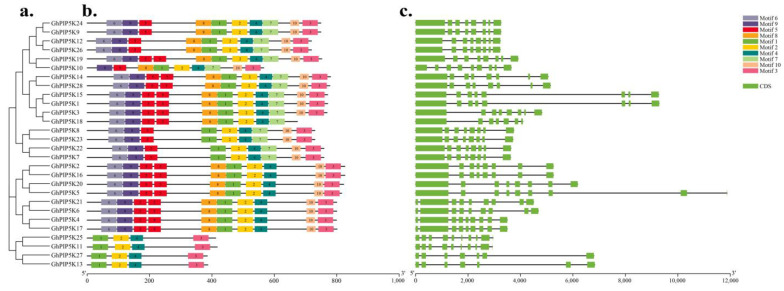
Comparison of the conserved motifs and gene structure of *GhPIP5Ks*. (**a**). Unrooted phylogenetic tree of *GhPIP5Ks*. (**b**). The arrangement of motifs in *G. hirsutum* was depicted using colored boxes, with each color representing a different number from 1 to 10. (**c**). A structure analysis revealed the exons and introns in *GhPIP5Ks*, which were represented by green boxes and black lines, respectively. A scale bar was included for reference at the bottom of the figure.

**Figure 5 ijms-25-01511-f005:**
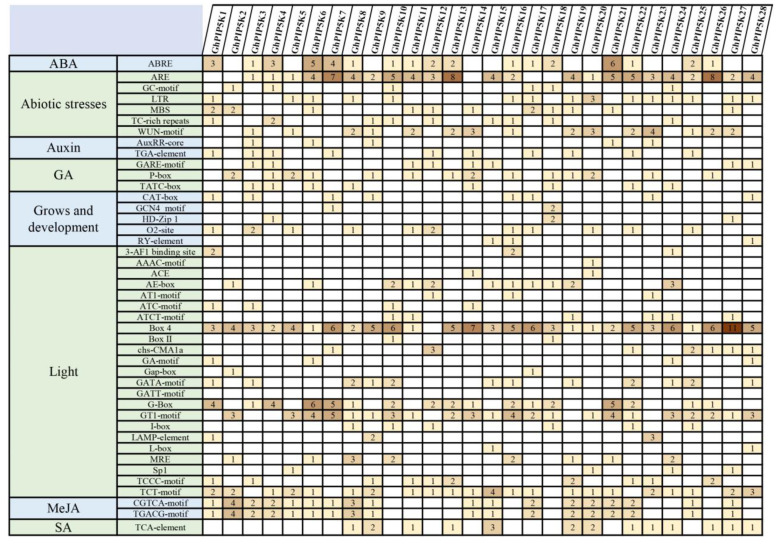
Projected results concerning cis-regulatory elements located in the promoter regions of *GhPIP5Ks*. The cell values represent the counts of regulatory elements. The intensity of the color increases with increase in the numerical value.

**Figure 6 ijms-25-01511-f006:**
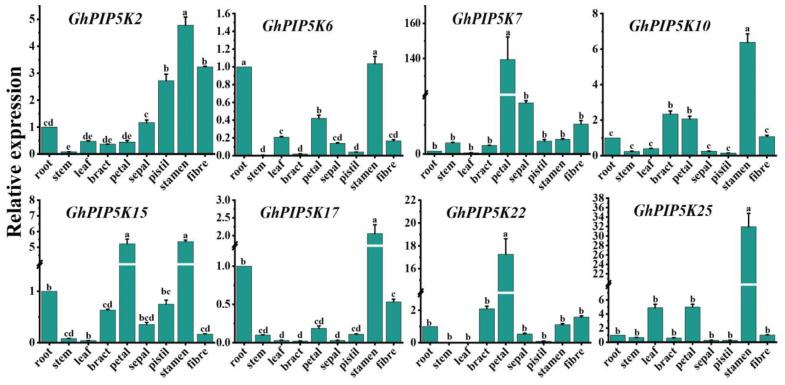
Expression analysis of *GhPIP5K* genes in different tissues of upland cotton under normal growth conditions. The experiment was conducted in triplicate, and the data were analyzed using Student’s *t*-test. The different letters (a, b, c, d and e) indicated significant differences based on Duncan’s Multiple Range test.

**Figure 7 ijms-25-01511-f007:**
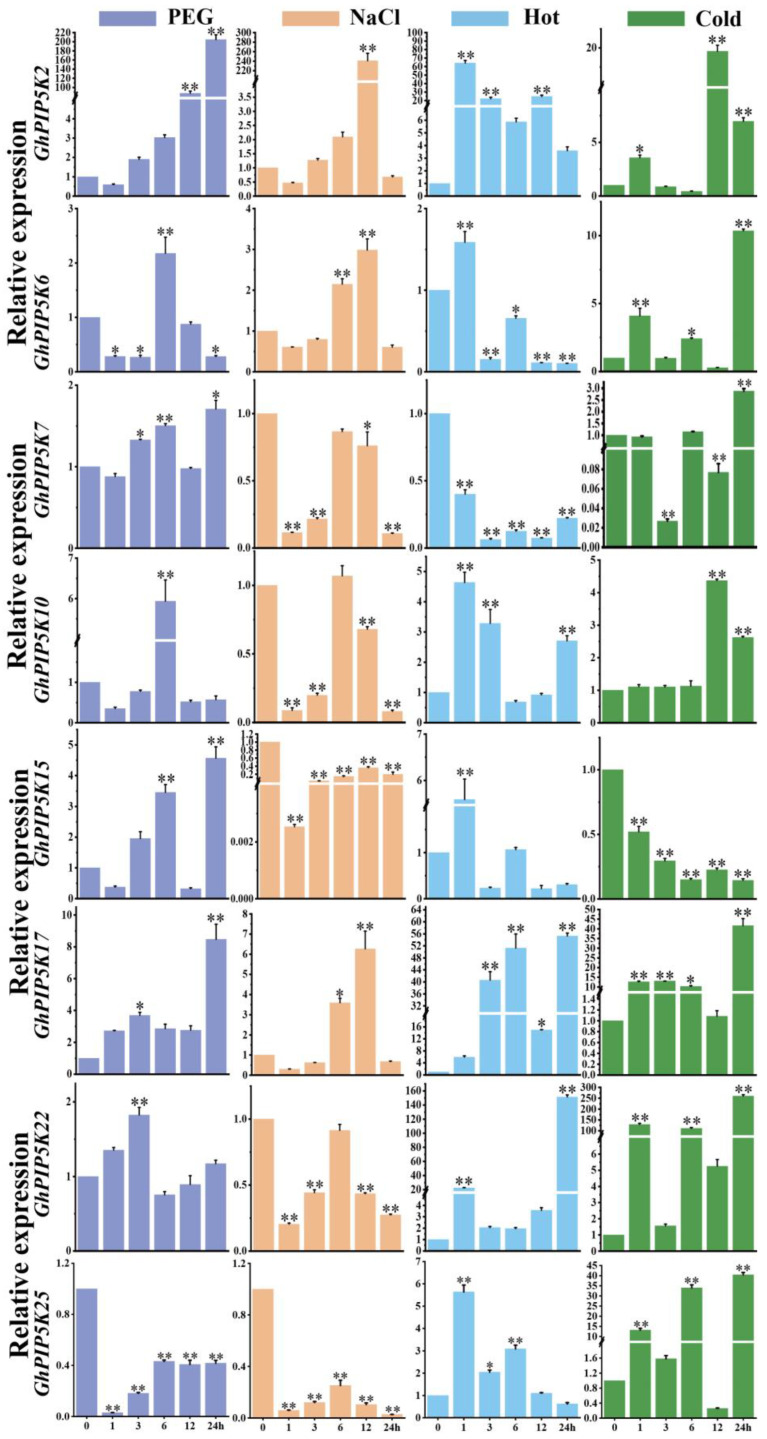
Relative expression levels of 8 *GhPIP5Ks* under diverse environmental conditions, including PEG, salt, high temperature, and low temperature stress. The error bars on the graph represent the standard deviations calculated from three replicates. The colors used in the graph indicate the different treatments: purple indicates drought treatment, orange indicates salt stress treatment, blue indicates high temperature treatment, and green indicates low temperature treatment. Asterisks denote a notable level of significance in relation to the control value (** p* < 0.05, *** p* < 0.01).

**Figure 8 ijms-25-01511-f008:**
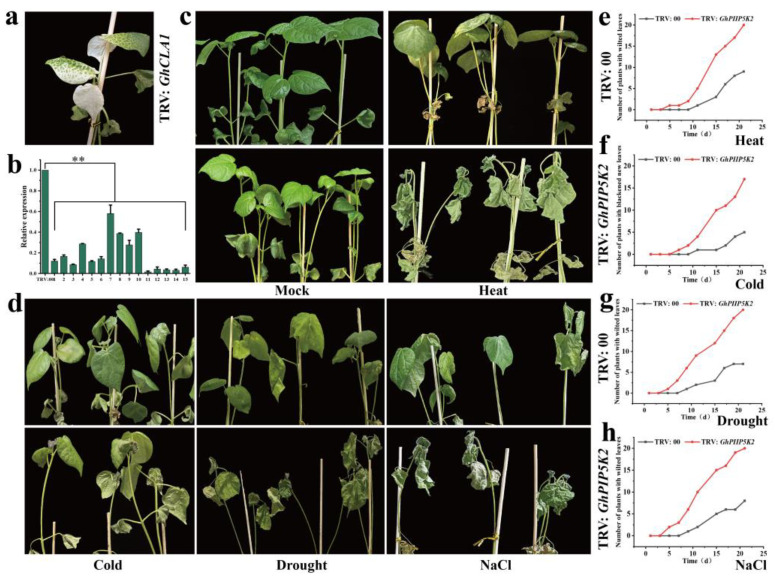
Validation of *GhPIP5K2* by the VIGS technique. (**a**). The leaves of TRV:*GhCLA1* cotton plants (positive control) exhibited an albino appearance. (**b**). Detection of the silencing efficiency of *GhPIP5K2* via qRT‒PCR. Asterisks denote a notable level of significance in relation to the control value (*** p* < 0.01). (**c**,**d**). Effects of heat, cold, drought and NaCl stress on the TRV:00 and TRV:*GhPIP5K2* phenotypes. (**e**). Numbers of fallen leaves from the control and silenced plants after exposure to heat stress. (**f**). Numbers of blackened leaves on the control and silenced plants after exposure to cold stress. (**g**). Numbers of fallen leaves from the control and silenced plants after exposure to drought stress. (**h**). Numbers of fallen leaves from the control and silenced plants after exposure to salt stress.

**Figure 9 ijms-25-01511-f009:**
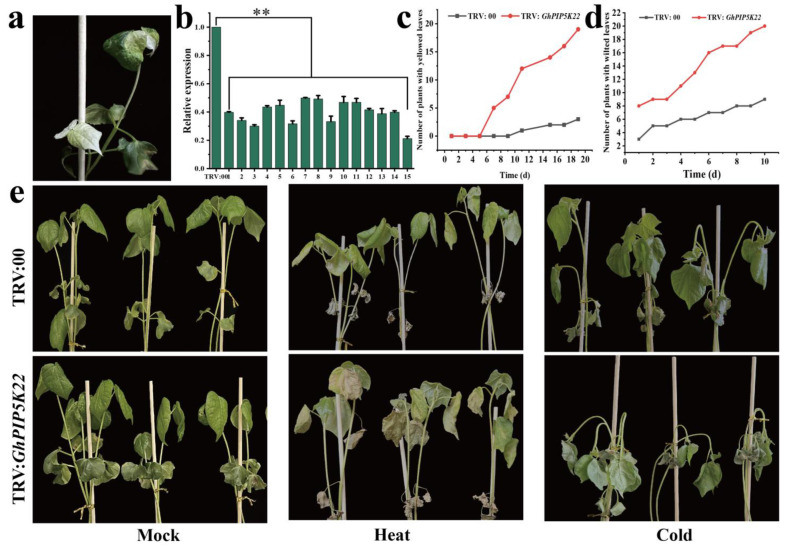
Validation of *GhPIP5K22* by the VIGS technique. (**a**). The leaves of the TRV:*GhCLA1* plants (positive control) exhibited an albino appearance. (**b**). Detection of the silencing efficiency of *GhPIP5K22* via qRT‒PCR. Asterisks denote a notable level of significance in relation to the control value (*** p* < 0.01). (**c**). Numbers of fallen leaves from the control and silenced plants after exposure to heat stress. (**d**). Numbers of wilted leaves on the control and silenced plants after exposure to cold stress. (**e**). Effects of heat and cold stress on the TRV:00 and TRV:*GhPIP5K2* phenotypes.

**Figure 10 ijms-25-01511-f010:**
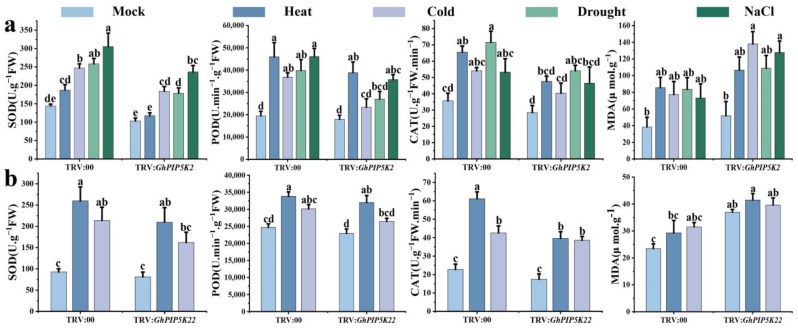
Detection of physiological and biochemical indices of TRV:00, TRV:*GhPIP5K2* and TRV:*GhPIP5K22* plants before and after stress treatment. (**a**). SOD activity, POD activity, CAT activity and MDA content of TRV:*GhPIP5K2*. (**b**). SOD activity, POD activity, CAT activity and MDA content of TRV:*GhPIP5K22*. The experiment was conducted in triplicate, and the data were analyzed using Student’s *t*-test. The different letters (a, b, c and d) indicated significant differences based on Duncan’s Multiple Range test.

**Figure 11 ijms-25-01511-f011:**
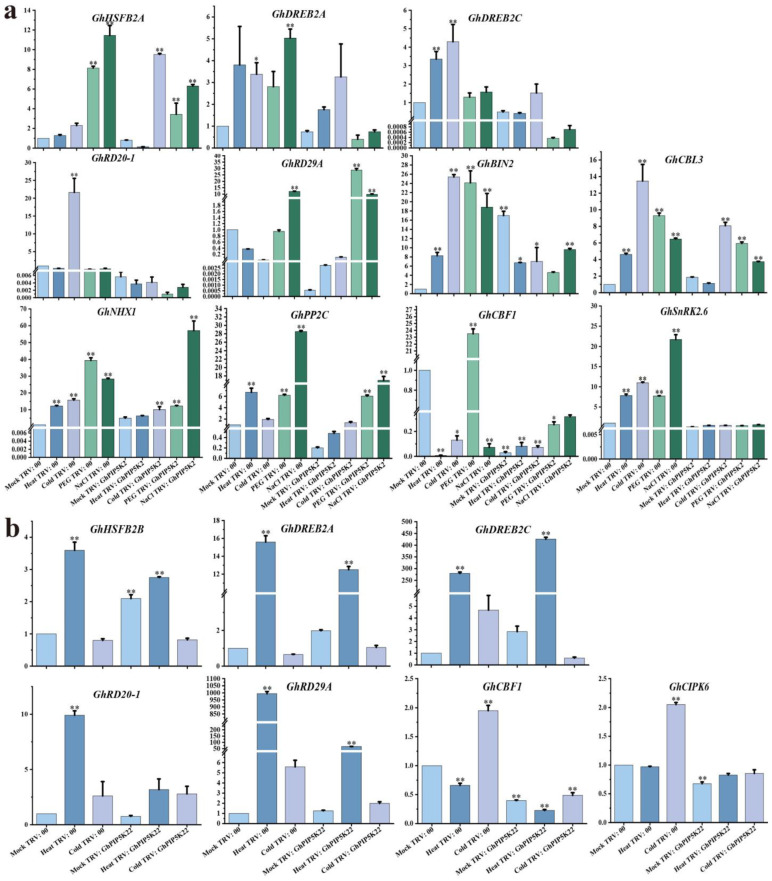
Changes in the expression of stress-related genes in TRV:00, TRV:*GhPIP5K2* and TRV:*GhPIP5K22* plants before and after stress treatments. (**a**). Expression of 11 stress marker genes in TRV:*GhPIP5K2*. (**b**). Expression of 7 stress marker genes in TRV:*GhPIP5K2*. Asterisks denote a notable level of significance in relation to the control value (* *p* < 0.05, ** *p* < 0.01).

**Table 1 ijms-25-01511-t001:** Subcellular distribution and physicochemical properties of *GhPIP5Ks*.

Sequence ID	Gene Name	Nm. of Amino Acids	Molecular Weight	Theoretical pI	Instability Index	Aliphatic Index	Grand Average of Hydropathicity	Subcellular Localization
*GH_A01G2098*	*GhPIP5K1*	770	88,582.97	8.59	31.89	63.94	−0.71	nucleus
*GH_A01G2293*	*GhPIP5K2*	825	93,092.51	8.93	43.57	76.76	−0.52	nucleus
*GH_A02G1290*	*GhPIP5K3*	768	88,787.07	6.05	36.25	64.35	−0.70	cytosol
*GH_A03G2365*	*GhPIP5K4*	798	90,509.07	9.06	39.79	71.03	−0.56	nucleus
*GH_A04G0885*	*GhPIP5K5*	815	92,004.38	8.77	47.62	78.65	−0.49	nucleus
*GH_A04G1543*	*GhPIP5K6*	799	90,517.74	8.79	37.85	69.42	−0.60	nucleus
*GH_A05G2320*	*GhPIP5K7*	758	86,497.71	8.50	31.74	64.79	−0.63	nucleus
*GH_A05G2701*	*GhPIP5K8*	729	83,795.41	8.01	34.20	63.50	−0.71	nucleus
*GH_A05G2841*	*GhPIP5K9*	748	85,400.57	9.02	37.94	59.79	−0.72	nucleus
*GH_A05G4205*	*GhPIP5K10*	565	64,445.81	7.57	32.24	63.27	−0.59	cytosol
*GH_A08G1524*	*GhPIP5K11*	416	47,724.25	8.83	53.92	76.13	−0.39	nucleus
*GH_A10G2027*	*GhPIP5K12*	718	81,362.62	7.80	28.89	59.97	−0.62	cytosol
*GH_A12G1261*	*GhPIP5K13*	386	44,712.66	8.47	38.77	86.58	−0.23	cytosol
*GH_A13G0020*	*GhPIP5K14*	779	89,410.94	8.37	35.55	66.59	−0.66	nucleus
*GH_D01G2192*	*GhPIP5K15*	770	88,601.99	8.41	32.68	64.31	−0.71	nucleus
*GH_D01G2372*	*GhPIP5K16*	825	93,052.47	8.93	42.74	76.06	−0.53	nucleus
*GH_D02G2535*	*GhPIP5K17*	800	90,634.33	9.16	39.92	69.51	−0.58	nucleus
*GH_D03G0675*	*GhPIP5K18*	673	77,580.07	5.83	36.86	62.29	−0.74	nucleus
*GH_D04G0173*	*GhPIP5K19*	751	85,098.08	6.72	36.26	61.49	−0.65	nucleus
*GH_D04G1204*	*GhPIP5K20*	821	92,894.35	8.90	49.49	76.88	−0.51	nucleus
*GH_D04G1886*	*GhPIP5K21*	799	90,552.9	8.92	39.15	70.40	−0.59	nucleus
*GH_D05G2342*	*GhPIP5K22*	758	86,343.47	8.41	29.17	64.27	−0.64	nucleus
*GH_D05G2718*	*GhPIP5K23*	729	83,625.2	8.18	34.31	62.96	−0.71	nucleus
*GH_D05G2857*	*GhPIP5K24*	748	85,445.65	9.02	38.33	59.79	−0.72	chloroplast
*GH_D08G1541*	*GhPIP5K25*	411	46,984.46	8.61	55.42	79.90	−0.39	nucleus
*GH_D10G2136*	*GhPIP5K26*	718	81,412.64	7.52	29.62	59.83	−0.63	cytosol
*GH_D12G1277*	*GhPIP5K27*	384	44,353.16	8.91	35.73	82.68	−0.25	nucleus
*GH_D13G0016*	*GhPIP5K28*	777	89,212.72	8.46	36.42	66.00	−0.67	nucleus

## Data Availability

All the data generated or analyzed during this study are included in this article and its Appendix A.

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
