# Peer review of "The Silencing of GhPIP5K2 and GhPIP5K22 Weakens Abiotic Stress Tolerance in Upland Cotton (Gossypium hirsutum)"

_ijms, 2024, doi:10.3390/ijms25031511_

Round 1

Reviewer 1 Report

Comments and Suggestions for Authors

The manuscript by Ling et al. entitled “Silencing of GhPIP5K2 and GhPIIP5K22 Weakens Abiotic Stress Tolerance in Upland Cotton (Gossypium hirsutum)” investigated Phosphatidylinositol 4-phosphate 5-kinases (PIP5Ks) by bioinformatics analysis and virus-induced gene silencing assay. The authors claim that ‘provide a foundation for further inquiry into the actions of the GhPIP5K gene family in regulating the response and resistance mechanisms to abiotic stresses in cotton’ based on the effectiveness of GhPIP5K2 and GhPIP5K22 on cotton plants. The work is technically sound piece of research, and within the scope of IJMS, it may need a major revision before its acceptance for publication.

1.       The grammar/logic issues in the text, for example: ‘Cotton (Gossypium spp.) is a globally important primary source of fiber in agriculture’,

2.       I doubt http://www.cottonfgd.org/ is a fake website

3.       The authors could provide the overexpression effectiveness data of GhPIP5K2 and GhPIP5K22 to add the manuscript weight

4.       MEGA 7.0 is an 8-year-old software

5.       For qPCR, it is not work well input 200 ng of cDNA as templates, to my best knowledge (maybe pg?)

6.       The sheet name should be same as the title of the supplemental tables

7.       The primers corresponding gene acc. num. should be listed on the Table S3

Comments on the Quality of English Language

need to be polished 

Author Response

Thank you for your helpful suggestions and the informative comments. We have studied the comments carefully and have made major revisions. All the comments were valuable, and provide important guiding significance for our research. We carefully revised the manuscript according to the comments, and marked the changes in red in the text. Because the images in the response letter cannot be copied directly to “Your Response” in the “Author Dashboard”, we are submitting our responses, including these images, to the editors and reviewers via a cover letter. The comments and our responses are as follows:

Question 1. The grammar/logic issues in the text, for example: ‘Cotton (Gossypium spp.) is a globally important primary source of fiber in agriculture’,

Answer 1. We apologize for the error. We have revised the sentence in line 86-87 on page 2 in of the article. To avoid such errors, the paper was sent to for American Journal Experts editing.

Question 2. I doubt http://www.cottonfgd.org/ is a fake website.

Answer 2. We apologize for the error. This website is commonly used for cotton, and the specific use and function of the CottonFGD website has previously been described in the following article: Zhu T, Liang C, Meng Z, Sun G, Meng Z, Guo S, Zhang R. CottonFGD: an integrated functional genomics database for cotton. BMC Plant Biol. 2017 Jun 8;17(1):101. doi: 10.1186/s12870-017-1039-x. PMID: 28595571; PMCID: PMC5465443. The same website was also used in the study described in the following article Dai M, Zhou N, Zhang Y, Zhang Y, Ni K, Wu Z, Liu L, Wang X, Chen Q. Genome-wide analysis of the SBT gene family involved in drought tolerance in cotton. Front Plant Sci. 2023 Jan 11;13:1097732. doi: 10.3389/fpls.2022.1097732. PMID: 36714777; PMCID: PMC9875013. However, due to the frequent updating and maintenance of this website, it will often not be able to open, We have modified the latest URL of this website as follows: https://cottonfgd.org/.

Question 3. The authors could provide the overexpression effectiveness data of GhPIP5K2 and GhPIP5K22 to add the manuscript weight.

Answer 3. Thank you for this constructive suggestion. Arabidopsis overexpression data can be used to validate the functions of the GhPIP5K2 and GhPIP5K22 genes, but during the cotton experiments, the duration of the T3 generation of overexpressing Arabidopsis thaliana plants was relatively long, and the experiments could not be completed in a short time. In the future, we will conduct experiments related to overexpression in Arabidopsis plants.

Question 4. MEGA 7.0 is an 8-year-old software.

Answer 4. Thank you for the good reminder. MEGA 7.0 is indeed an old software package, so we have reverted to visualizing the data using MEGA 11.0.13 software. The same diagram was obtained with both software.

Question 5. For qPCR, it is not work well input 200 ng of cDNA as templates, to my best knowledge (maybe pg?).

Answer 5. For qPCR, our laboratory utilized a kit obtained from Tengen (item number FP209). Initially, the cDNA concentration in ng/µl was determined using a NanoDrop 2000 spectrophotometer (Thermo Scientific, USA). Subsequently, the concentration was diluted to 100 ng/µl, and 2 µl was subsequently added to the qPCR system. However, we will follow your suggestion and test a template concentration gradient in subsequent experiments to identify the optimal concentration. According to the literature [Wu P, Lu C, Wang B, Zhang F, Shi L, Xu Y, Chen A, Si H, Su J, Wu J. Cotton RSG2 Mediates Plant Resistance against Verticillium dahliae by miR482b Regulation. Biology (Basel). 2023 Jun 23;12(7):898. doi: 10.3390/biology12070898. PMID: 37508331; PMCID: PMC10376429.], 1 to 2 µl of cDNA was added to the qPCR system.

Question 6. The sheet name should be same as the title of the supplemental tables.

Answer 6. We apologize for the error and have made changes to the Supplementary Tables.

Question 7. The primers corresponding gene acc. num. should be listed on the Table S3.

Answer 7. Thank you for the good suggestion. We have added this information to Table S3 in the supplementary materials.

Reviewer 2 Report

Comments and Suggestions for Authors

The submitted manuscript to IJMS entitled “Silencing of GhPIP5K2 and GhPIIP5K22 Weakens Abiotic Stress Tolerance in Upland Cotton (Gossypium hirsutum)” is of great potential to be published. But, before publication, following are the comments that need to be addressed:

How were the different stresses employed to the plants? Please explain it in the M*M section. Without knowing about this information, the reviewer is unable to find a way to compare the findings!

Why is it worthful to investigate the effect of low temperature on cotton? Is it really a problem under field conditions?

The quality of figures is not appropriate.

Figure 6: Expression levels of which treatment are shown?

Please always mention the change (increase or decrease) in percentage or fold-change where it is applicable.

Please mention the significant letters rather than asterisk where applicable.

Figure 10: Why there was no drought and NaCl treatment for TRV:GhPIP5K22?

The conclusion section should be revised.

Comments on the Quality of English Language

Minor revision

Author Response

Thank you for your helpful suggestions and the informative comments. We have studied the comments carefully and have made major revisions. All the comments were valuable, and provide important guiding significance for our research. We carefully revised the manuscript according to the comments, and marked the changes in red in the text. Because the images in the response letter cannot be copied directly to “Your Response” in the “Author Dashboard”, we are submitting our responses, including these images, to the editors and reviewers via a cover letter. The comments and our responses are as follows:

Question 1.  How were the different stresses employed to the plants? Please explain it in the M*M section. Without knowing about this information, the reviewer is unable to find a way to compare the findings!

Answer 1. Thank you for the good suggestion. We have added a description of the experimental materials and treatments to section 4.6 of the Materials and Methods on lines 543-556 in page 20.

Question 2.  Why is it worthful to investigate the effect of low temperature on cotton? Is it really a problem under field conditions?

Answer 2. Thank you for the good question. Low-temperature cold injury mainly occurred during the sowing, emergence, seedling and later growth stages, resulting in low-temperature seed rot, bud rot, root rot and insufficient development of cotton bolls, all of which affect the cotton yield and quality. Moreover, due to the gradual increases in the global greenhouse effects, extreme temperature events such as cold waves occur frequently [Devireddy AR, Zandalinas SI, Fichman Y, Mittler R. Integration of reactive oxygen species and hormone signaling during abiotic stress. Plant J. 2021;105:459–76. doi: 10.1111/tpj.15010], and the probability that crops will suffer from low-temperature stress will become increasingly severe in the 21st century [Kodra E, Steinhaeuser K, Ganguly AR. Persisting cold extremes under 21st-century warming scenarios. Geophys Res Lett. 2011;38(8):8705. doi: 10.1029/2011GL047103]. The following article, mentioneds that cotton may be affected by low temperature and cold injury at various growth stages, and these results suggest that low-temperature stress can negatively impact the expression of insecticidal proteins in Bt transgenic cotton [Liu Z, Ji M, He R, Dai Y, Liu Y, Mou N, Du J, Zhang X, Chen D, Chen Y. Effect of  Low Temperature on Insecticidal Protein Contents of Cotton (Gossypium herbaceum L.) in the Boll Shell and Its Physiological Mechanism. Plants (Basel). 2023 Apr 26;12(9):1767. doi: 10.3390/plants12091767. PMID: 37176825; PMCID: PMC10180954].

Question 3.  The quality of figures is not appropriate.

Answer 3. Thank you for this constructive suggestion. We have increased the resolution of all the images.

Question 4. Figure 6: Expression levels of which treatment are shown?

Answer 4. Thank you for the good question. Figure 6 shows the results from the analysis of the expression of GhPIP5K gene members in different tissues of upland cotton under normal growth conditions. We have made changes to the text on lines 263-266 in page 10.

Question 5.  Please always mention the change (increase or decrease) in percentage or fold-change where it is applicable.

Answer 5. Thank you for the good reminder. We have revised the Results section on lines 276-297 in pages 11 and 12.

Question 6.  Please mention the significant letters rather than asterisk where applicable.

Answer 6. Thank you for this constructive suggestion. We have made the following modifications on lines 263-266 in page 10 and on lines 371-373 in page 15 of the revised manuscript: “The experiment was conducted in triplicate, and the data were analyzed using Student’s t test. The different letters (a, b, c, d and e) indicate significant differences based on Duncan’s honestly significant differences test”.

Question 7.  Figure 10: Why there was no drought and NaCl treatment for TRV:GhPIP5K22?

Answer 7. Thank you for this reasonable question. This information was mentioned in section 2.7. The following text was included on lines 288-293 in page 11: “the expression of GhPIP5K2 increased hundreds of times under drought, NaCl and heat stress and tens of times under cold stress. According to the qRT‒PCR results, the expression of GhPIP5K22 did not change significantly under drought or salt stress; thus, only heat and cold stress were applied.

Question 8.  The conclusion section should be revised.

Answer 8. Thank you for the good reminder. The main changes made to the Conclusion section on lines 319-321 in page 13 of the revised manuscript are as follows: “These results indicated that GhPIP5K2 had a positive effect on the responses of cotton to abiotic stress, whereas GhPIP5K22 had a positive effect on the responses to heat and cold conditions”.